# National Holidays and Social Mobility Behaviors: Alternatives for Forecasting COVID-19 Deaths in Brazil

**DOI:** 10.3390/ijerph182111595

**Published:** 2021-11-04

**Authors:** Dunfrey Pires Aragão, Davi Henrique dos Santos, Adriano Mondini, Luiz Marcos Garcia Gonçalves

**Affiliations:** 1Pós-Graduação em Engenharia Elétrica e de Computação, Universidade Federal do Rio Grande do Norte, Av. Salgado Filho, 3000, Lagoa Nova, Natal 59078-970, Brazil; dunfrey.aragao.073@ufrn.edu.br (D.P.A.); davihenriqueds@gmail.com (D.H.d.S.); 2Faculdade de Ciências Farmacêuticas, Universidade Estadual Paulista “Júlio Mesquita Filho”, Rodovia Araraquara-Jaú, Km 1, Campus Ville, Araraquara 14800-903, Brazil; adriano.mondini@unesp.br

**Keywords:** COVID-19, epidemiological SEIRD model, PCA, LSTM, time-series forecast

## Abstract

In this paper, we investigate the influence of holidays and community mobility on the transmission rate and death count of COVID-19 in Brazil. We identify national holidays and hallmark holidays to assess their effect on disease reports of confirmed cases and deaths. First, we use a one-variate model with the number of infected people as input data to forecast the number of deaths. This simple model is compared with a more robust deep learning multi-variate model that uses mobility and transmission rates ($R_{0}$, $R_{e}$) from a SEIRD model as input data. A principal components model of community mobility, generated by the principal component analysis (PCA) method, is added to improve the input features for the multi-variate model. The deep learning model architecture is an LSTM stacked layer combined with a dense layer to regress daily deaths caused by COVID-19. The multi-variate model incremented with engineered input features can enhance the forecast performance by up to 18.99% compared to the standard one-variate data-driven model.

## 1. Introduction

COVID-19 is a serious acute respiratory syndrome caused by the beta-coronavirus SARS-CoV-2, which was first reported at the end of 2019 [1]. The rate of transmission has ranked COVID-19 as the worst pandemic of the century in terms of scale and speed [2]. Transmission can occur through direct, indirect, or close contact with secretions of infected individuals or through direct contact with infected surfaces. SARS-CoV-2 enters the host cells via interaction with its entry receptor, angiotensin-converting enzyme 2 (ACE2), and an activating receptor, a protease such as TMPRSS2 or cathepsin [3].

The first case of COVID-19 in South America was reported on 25 February 2020 in the city of São Paulo, Brazil, which is an important travel hub for the region [4]. Since then, important control measures, such as overall or partial closing of marine, land, and air borders; travel restrictions, shutdown of schools and colleges; and imposed lockdown were implemented in different ways in Brazil and other countries of the region. According to official data, the country has reported the highest number of cases of COVID-19 in South America [5,6].

Brazil has the highest Gross Domestic Product (GDP) in South America, and the population density varies according to the regional division of the country. The mean population density is 24.69 inhabitants per square kilometer [7]. According to Organisation for Economic Co-operation and Development (OECD) [8], Brazil is composed of approximately 1.3 million independent or liberal professionals. Informal employment ranges from 20 to 49% of the workforce of the country. Formal and informal work positions were directly affected by restrictions to contain the spread of the virus, most of them based on social distancing. Data collected from 239 slum communities [9], where approximately 6.5% of the Brazilian population lives, showcase part of the Brazilian reality for formal and informal workers during COVID-19 pandemic. Approximately 72% of residents report that they do not have any savings to fall back on, while 15% had only the equivalent of one minimum wage in savings to survive the next month. Approximately 50% of the residents of these communities are liberal professionals or rely on informal work positions as their main source of income.

National and hallmark holidays usually involve a massive mobility of people seeking stores and malls, parks, and beaches. During the COVID-19 pandemic, the 2020 Brazilian calendar maintained nine days of national holidays; seven of them included extended weekends, from Friday to Sunday and/or Monday, which increases the circulation of individuals. At the beginning of February, crowds were reported across the country [10]. The government of the city of São Paulo estimates that the local carnival took about 15 million people to the streets in February of 2020 [11].

To date, the Brazilian scenario has demonstrated that the pandemic just deepened the already-existing political, social, and economic issues in the country [7]. Although the universal Brazilian public health system has been an example to the world struggling to manage other major outbreaks such as dengue fever, measles, Zika, and chikungunya viruses [12,13], the country has been coping with the several issues that affect COVID-19 prevention. The main ones are the lack of water supply, limited access to hand sanitizers and masks, and the lack of community engagement [7]. Social distance, among some 6068 non-pharmaceutical interventions, is the most effective method adopted by leaders around the world, presenting a major impact on decreasing transmission rate $R_{e}$ [14]. As it turns out, the dilemma to adopt social distance in socioeconomically disadvantaged areas such as slums is difficult to assess [15,16].

In this context, this study has three main contributions: (1) investigating the impact of Brazilian national holidays in social distancing and the evolution of COVID-19 in the country; (2) assessing the reproduction number using SEIRD model as well as applying Principal Component Analysis (PCA) to reduce the dimensions of a dataset containing community mobility reports, using the outcomes produced by these methods as input data for regression; and (3) demonstrating that estimates from data-driven pipelines based on holidays and using multi-variate LSTM neural networks are appropriate to predict 14-day COVID-19 daily deaths in Brazil. Our study provides evidence for the impact of holidays, community mobility, and the association of these factors with crowding. Our results indicate that an acceleration of the spread of the virus happens after the holiday breaks, which may eventually influence the access to adequate medical attention or ICU beds.

## 2. Materials and Methods

The open-access dataset of COVID-19 consists of official reports provided by the sanitary authority of Brazilian states [17]. The dataset contains country-level, daily-updated data retrieved from the Brazilian Ministry of Health and Brazilian Institute of Geography and Statistics (IBGE). The dataset contains reported cases, daily fatalities, number of cases per epidemiological week, number of deaths per day, total number of deaths, and reports of COVID-19 recovered and vaccinated individuals among others (Table 1). Data are presented in absolute numbers or in percentage per 100,000 inhabitants. Our analysis contains a time series that started in 25 February 2020, comprising features for all 26 Brazilian states.

SARS-CoV-2 dispersed throughout the country rapidly after the first official report. On 13 April 2021, the 16th week of the year, 82,186 cases of COVID-19 and 3808 daily deaths were reported in Brazil. Considering a 7-day moving average, these numbers represent approximately 71,344 cases and 3068 deaths (Figure 1a,b). These values are the result of the sum of reports in all Brazilian states. Nevertheless, a certain stability is observed in the death curve in the period ranging from weeks 22 to 34 (Figure 1b). This could be attributed to several factors, including non-pharmaceutical interventions adopted to reduce the contact among people, which ultimately influence the amount of viral load that an individual is exposed to. Additionally, the relative availability of hospitals that were not operating at their full capacity provided proper access to ICU beds and adequate management of infected patients.

Geographic distributions for deaths by COVID-19 in all Brazilian states are shown in Figure 2. The majority of casualties are concentrated in the Southeast region of the country, which includes the states of Espírito Santo (ES), São Paulo (SP), Rio de Janeiro (RJ), and Minas Gerais (MG).

Instead of using data from individual Brazilian states in the analysis, the main contribution of this work is to understand the social component across the country, despite the effect of local singularities in each Brazilian city and state. Therefore, we selected three populated states with the highest mortality rates for COVID-19 in Brazil to briefly highlight such singularities: São Paulo, Rio de Janeiro, and Minas Gerais. These states have 40% of the estimated Brazilian population.

Population densities are a result of the size of each state. São Paulo, for example, has an estimated population of 46,649,132 inhabitants and a density of 166.25 inhab/km${}^{2}$. People are relatively dispersed in the state. However, more than 22 million people are concentrated on the metropolitan area of the city of São Paulo. The same happens to Minas Gerais, with a population of approximately 21.5 million people, density of 33.41 inhab/km${}^{2}$, and more than six million people residing on the metropolitan area of the state capital, Belo Horizonte. The population density of the third smallest state in the country is 365.23 inhab/km${}^{2}$, and like the other two states, approximately 76% of the 17,463,349 inhabitants live in the metropolitan area of the state capital, the city of Rio de Janeiro [18]. Altogether, the three states have a major influence on Brazilian reports of COVID-19, but São Paulo seems to help shape the curve of Brazilian daily cases (Figure 3).

The Brazilian official calendar includes several federal holidays and religious festivities that can lead to crowding. State holidays and regional festivities were not included in the analysis. There were 12 official national holidays for 2020 (Table 2). Due to the severity of COVID-19 worldwide, the Brazilian Ministry of Defense followed recommendations of health authorities and canceled military parades and other festivities of September 7th to avoid public events that could increase the spread of new variants of SARS-CoV-2 [9,19]. The two rounds of Brazilian elections, in November 15th and in November 29th, were also included as potential sources of crowding.

Holidays usually involve increased mobility of people. Thus, we assessed mobility of Brazilian community using data retrieved from Google LLC (Alphabet Inc., Mountain View, CA, USA) during the epidemiological weeks evaluated herein. The company started publishing mobility reports [20] in early April 2020. These reports showcase how COVID-19 and countermeasures influenced the dynamics of mobility over time. Mobility database presents median of data collected daily after removal of noises rather than raw quantities [21].

Changes and trends observed in mobility data in several contexts, such as retail and recreation, supermarket and pharmacy, parks, transit stations, businesses, and residential areas, and the variation of average daily cases of COVID-19 per week can be seen in Figure 4. There is a contrast in mobility patterns, daily reports, and deaths from weeks 14 to 20 of 2021 (Figure 1b and Figure 4b). The increase in the number of cases was followed by a growth of death reports during the worst chapter of COVID-19 pandemic in Brazil. As a result, celebrations were canceled and stricter measures to contain mobility were implemented. Thus, the assessment of holidays and mobility patterns can reveal trends to be used as input data in regression systems to study COVID-19.

All of the figures, codes, and tests used on the graphics above were written in Python 3.7. Software libraries Pandas 1.3, TensorFlow v2.6.0, NumPy 1.21, Matplotlib stable release 3.4.3, and the free and open-source Python library SciPy 1.21.2 were used.

### 2.1. Principal Component Analysis

The mobility report dataset provided by Google contains a complex arrangement of information and patterns that are influenced by different community sectors, habits, and external events. Nevertheless, with the transformation of such information into data, it is possible to produce an independent variable that may be able to generalize a temporal event or situation. Principal Component Analysis (PCA) allowed us to perform the analysis of different projections of the dataset without losing significant information and to generalize dimensions with a smaller number of data.

The PCA method reduces the number of dimensions in a scenario with multiple variables by using a linear transformation to turn a large number of correlated original variables into a smaller number of uncorrelated variables [22]. The newly discovered dimensions are smaller or equal in size to the original variables used in the algorithm [23].

PCA is a statistical multi-variate analysis method that aims to identify the main factors that cause the most variation in a set of data. As a result, we consider as the primary goal the information contained in several original variables in a smaller set with as little information loss as possible.

To set up the sorting of the principal components (PC) of mobility reports data using PCA, the first step was to measure the average value of all dimensions of the database. To avoid unequal supplying of contribution for a given dimension into the process, we applied a method to standardize the data to generate input in a common scale. This mapping is given by:(1)$$z=\frac{x-\mu }{\sigma }$$
where *z* is the scaled value, *x* is the original value, and μ and σ are the mean and standard deviation, respectively.

Next, we computed the covariance of variables and established the covariance matrix ***A***, which is given by:(2)$$cov\left(x,y\right)=\frac{1}{n-1}\sum_{i=1}^{n}\left(x_{i}-\bar{x}\right)\left(y_{i}-\bar{y}\right)$$

When a linear transformation is applied to a nonzero vector, the *eigenvector*, or characteristic vector of that linear transformation, changes by a scalar factor. The factor by which the *eigenvector* is scaled corresponds to the *eigenvalue*. Due to this transformation, it is necessary to compute *eigenvectors* and corresponding *eigenvalues* of the matrix ***A***:$$A\vec{v}=\lambda \vec{v}$$
$$A\vec{v}-\lambda \vec{v}=0$$
(3)$$\vec{v}\left(A-I\lambda \right)=0$$
where I is the identity matrix, and λ is the *eigenvalue*, which is a scalar value. This means that λ defines the linear transformation. Finally, the *eigenvectors* are sorted by decreasing *eigenvalues*, and the *k* *eigenvectors* with the largest *eigenvalues* are chosen as the PC.

### 2.2. Epidemiological SEIRD Model

The *Susceptible–Exposed–Infected–Recovered–Dead* model, also known as SEIRD [24,25], is an epidemiological analysis based on the *Susceptible–Infected–Removed* model, known as SIR [26,27]. SEIRD is intended to enhance the SIR model in order to explain the evolution of a population in terms of transmissibility, contact rates, and the expected duration of infection in the course of an outbreak. Some assumptions must be made in SEIRD modeling:Population size (*N*) is constant;Demographic features are not implemented or adopted;Heterogeneity: an infected individual has an equal chance of contacting a susceptible person.

SEIRD categorizes the population in groups to analyze data and design forecasts based on reported cases:Susceptible (S): Individual who is prone to be infected on day *t*, and has never been infected and is not immune to infection;Exposed (E): Individual who has been exposed to the disease but was not able to infect another person nor show symptoms;Infected (I): Individual who is infected and producing virus that can potentially infect other individuals;Recovered (R): Individual who was ill and recovered on day *t* with alleged acquired immunity;Dead (D): Individual who died because of the infection.

The movement of individuals from one group to the other during the course of the outbreak is resolved by using Ordinary Differential Equations (ODE), which constitutes the dynamic of the SEIRD model. ODE are defined as follows:(4)$$\frac{dS}{dt}=-\frac{S}{N}\left(\beta I+\epsilon E\right)$$
(5)$$\frac{dE}{dt}=\frac{S}{N}\left(\beta I+\epsilon E\right)-\alpha E$$
(6)$$\frac{dI}{dt}=\alpha E-I\left(\gamma +\mu \right)$$
(7)$$\frac{dR}{dt}=\gamma I$$
(8)$$\frac{dD}{dt}=\mu I$$
where the infectious rate *beta* (β) represents infections per exposure, i.e., a susceptible individual has contact with an infected individual and presents a latent infection after exposure; the infectious rate *epsilon* (ϵ) represents the potential rate of infection per exposure, i.e., a susceptible individual that has mutual contact with an exposed/infected individual and may infect another susceptible individual; the transitional rate *alpha* (α) of an exposed individual to infect others is the average latent period $\frac{1}{\alpha }$. *Gamma* (γ) and *mu* (μ) represent, respectively, the recovery rate and rate at which infected people become deceased; $\frac{1}{\gamma }$ is the mean infectious period. Figure 5 synthesizes the model.

SIR was the first epidemiological model to use compartments. In this model, the whole population can be assigned to only three basic compartments, namely susceptible, infected, and removed. The last compartment includes people who recovered from the infection and the ones who died because of the infection. This is one of the limitations of the model. Disregarding exposure and the incubation period is another pitfall of using SIR analysis. An alternative approach to analyze the population in compartments during an outbreak is to employ the *Susceptible–Exposed–Infected–Removed–Susceptible* (SEIRS) model [28]. Unlike with SIR, individuals from the removed compartment can return to the susceptible compartment. However, our database provided death figures but lacked reinfection information. Thus, the SEIRD model proved to be more appropriate for our data set.

#### Basic Reproduction Number

The estimated number of secondary cases created by a single (typical) infection in a fully susceptible population is known as the basic reproduction number $R_{0}$, a dimensionless number, and is referred to in many cases as the *simple reproductive rate*.

A possible way to find $R_{0}$ is by adopting the next-generating matrix to calculate the reproduction number. Furthermore, the method calculates the current reproduction number $R_{e}$, which is the secondary infection rate. The model is divided into two groups: $X_{1}$, containing *E* and *I* individuals as the infective and infectious group, and $X_{2}$, which contains S, R, and D individuals as the susceptible, recovered, and dead group. Assuming S≈N and $f(X_{1},X_{2})$ and $v(X_{1},X_{2})$ are the vectors of new infection parameters and other parameters, respectively, then
(9)$$f\left(X,Y\right)=\left\{\begin{array}{c} \beta I+\epsilon E\\ 0 \end{array}\right\},\;\mathrm{and}\;v\left(X,Y\right)=\left\{\begin{array}{c} \alpha E\\ -\alpha E+I(\gamma +\mu ) \end{array}\right\}\;F=\frac{df}{dX}=\left[\begin{array}{cc} \epsilon & \beta \\ 0 & 0 \end{array}\right],\;\mathrm{and}\;V=\frac{dv}{dX}=\left[\begin{array}{cc} \alpha & 0\\ -\alpha & \gamma +\mu \end{array}\right]\;FV^{-1}=\left[\begin{array}{cc} \frac{\epsilon }{\alpha }+\frac{\beta }{\gamma +\mu } & \frac{\beta }{\gamma +\mu }\\ \; & \\ 0 & 0 \end{array}\right]$$

The *eigenvalue* of $FV^{-1}$ with the highest value is $R_{0}=\frac{\epsilon }{\alpha }+\frac{\beta }{\gamma +\mu }$ and $R_{e}=\frac{\beta }{\gamma +\mu }$. An outbreak is not likely to happen, or it is controlled, when $R_{0}\leq 1$. When $R_{0}>1$, however, the disease spreads exponentially, resulting in an epidemic.

### 2.3. Multi-Variate LSTM Model

Neural networks are useful tools for pattern recognition. Methods such as ARIMA [29,30] have been demonstrated to be insufficient in the long run to generalize or maintain accuracy of patterns in regressions of long periods. Some approaches are appropriate for problems that cannot be solved linearly. In such cases, the neural network allows us to identify the degree of relationships among variables, such as the ones in our study, since non-linear variables cannot have causality or correlation explained by commonly used methods. Thus, neural networks produce promising results for one-variate, bi-variate, and multi-variate regression problems.

In Long Short-Term Memory (LSTM) [31], similar to Recurrent Neural Network (RNN), a context of memory persists within the pipeline, allowing them to solve sequential and temporal subjects without being hampered by the vanishing gradient. These neural networks are built on the usage of gates that direct how information is forwarded and ignored inside its internal structures to achieve such complex learning retrieval from sequential sources.

LSTM differs from the traditional approach in that it contains an element called cell state, which determines whether the information is stored or not. The cell state can transport pertinent information throughout the sequence’s processing, cross the entire thread of interactions, add or delete data from this state cell, and set it according to structured switches.

Given this benefit, the goal is to use stacked LSTM units as a layer, composed of 200 blocks attached to a dense layer to predict a bi-weekly series of COVID-19 daily deaths taking into account one- and multi-variate input data, as shown in Figure 6.

For the one-variate model, we use COVID-19 daily cases as input data to predict the next 14 days of COVID-19 deaths. The learning algorithm is asked to output a function $f:R^{n}\to R$ in order to complete this task. For the multi-variate model, we set a mix of the data, finding the best approach, such as factor $R_{0}$ and/or principal components from PCA, to a multi-variate model to predict the next 14 days of COVID-19 deaths. The proposed model has the same architecture for any case, changing only its input data.

### 2.4. RMSE and Model Grid-Search

To evaluate the model, we apply the use of RMSE (root mean squared error), a metric that calculates the deviation of errors between observed (ground-truth) and predicted values (hypotheses):(10)$$RMSE=\sqrt{\frac{1}{n}\sum_{i=1}^{n}{\left(y_{i}-{\hat{y}}_{i}\right)}^{2}}$$
where $y_{i}$ is the true value, the ground truth, and ${\hat{y}}_{i}$ is the predicted value from the model. The sum will be given different weights and the RMSE index will rise dramatically as the instances’ error values rise. That is, if there is an outlier in the dataset, its weight in the RMSE calculation will be higher, harming the performance by increasing the RMSE.

Furthermore, a grid search to find the best set of input data for the model in forecasting the COVID-19 deaths with the lowest RMSE was applied. We apply the same experiment for each input dataset over 50 times, which allows us to extract the expected value and the standard deviation of each configuration in predicting our goal. For this, we split the data into training and testing data. The test set is related to the last 28 days (4 weeks) of the temporal series. Therefore, the RMSE extracted and reported in this work is related to comparisons against the test data.

The aforementioned workflow is described in Figure 7. The model receives each set of input data, and the model’s expected outcome applied to the test data is subjected to RMSE calculations. After the process has been repeated 50 times, we distill a boxplot graph to observe which configuration performs better on average, observing the standard deviation and mean RMSE calculated. To create a temporal plot, comparing the ground truth (real values) and the model’s predictions, we choose the model with the lowest RMSE in forecasting COVID-19 deaths for each set of input configuration. For example, for all 50 repetitions using cases as input data, we plot the curve with the lowest RMSE.

## 3. Experiments and Results

We performed a series of experiments in order to better understand whether community mobility on a public holiday could be a relevant element for predicting COVID-19 daily cases and deaths in Brazil. Initially, the basic reproduction numbers $R_{0}$ and $R_{e}$ were extracted from the SEIRD model, providing the association of national holidays and daily infections and deaths by COVID-19. Subsequently, PCA was applied for dimensionality reduction on the Community Mobility Report dataset. The dimensionless factors $R_{0}$, $R_{e}$, and the PCs generated by PCA were applied to the deep learning model as input data for the training step.

### 3.1. SEIRD Model to Extract Basic and Current Reproduction Rate ($R_{0}$ and $R_{e}$)

Based on the assumption that infection begins with a single person, the initial values of infected, $I_{initial}$, and exposed, $E_{initial}$, are set to 1. The initial values of the recovered, $R_{initial}$ (which we added the vaccinated individuals), and deceased, $D_{initial}$, groups are set to 0. To determine the initial value of susceptible $S_{initial}$, we use the following equation:(11)$$S_{initial}=N-E_{initial}-I_{initial}-R_{initial}-D_{initial}$$
where *N* is the population size or the total number of individuals. The incubation period for SARS-CoV-2 ranges from three to six days. During this period, infected individuals do not present symptoms [32,33]. Furthermore, COVID-19 symptoms, when present, usually appear up to 14 days after infection [32]. The SEIRD model uses parameters β, ϵ, α, γ, and μ to tune the forecast. Such parameters have been estimated by minimizing a function based on the least square error to obtain optimized values and an optimal solution.

We defined the initial values, lower bound, and upper bound as inputs for each parameter to recover their optimal value. Initial values and respective bounds are described as follows: β with the initial value 0.5 and bounds [0.001, 1]; ϵ with the initial value 0.001 and bounds [0.001, 1]; α with the initial value 1/4 and bounds [1/6, 1/2]; γ with the initial value 1/14 and bounds [1/18, 1/10]; and μ with the initial value 0.001 and bounds [0.001, 1].

To generate the periodic data containing $R_{0}$ and $R_{e}$, the pipeline was submitted to a loop process that consider a step of a day and take into account a period of 14 days (bi-weekly period). The extracted values are shown in Figure 8. The holidays and Brazilian elections over the COVID-19 daily cases curve are also shown in Figure 8.

We noticed a series of events that triggered crowding and, somehow, had an impact on Brazilian COVID-19 reports. Although the 7th of September holiday was canceled in 2020, there was an increase in COVID-19 reports. The same happened on All Souls’ Day, on 2 November. The trend in the curve of cases and the basic and current reproduction rates, $R_{0}$ and $R_{e}$, was probably influenced by proximity of people and infection caused by crowding with poor social distancing. Illegal crowds, which were not allowed at the time, may have also contributed to the increase of infections. The increase in reports was observed in each state and in the whole country (Figure 3).

From a period of 448 days of COVID-19 reports, 80 fell into a 14-day window after the holiday, as seen in Table 3. These windows, which we called holiday periods, represented 17.86% of the days in our analysis. Throughout holiday periods, $R_{0}$ values were usually above 1. Regular days, or the ones outside holiday periods, accounted for approximately 82% of the period of our study. $R_{0}$ was above one in only 23.21% of the regular days. As COVID-19 was officially reported in March of 2020 in Brazil, $R_{0}$ values were high at that point of the pandemic. We decided to exclude this month from the analysis. In this scenario, 15.94% of the days were holiday periods presenting $R_{0}>1$. Only 20% of the regular days had $R_{0}$ above one.

When *R*${}_{0}$ component is greater than 1, deviation is higher than when its value is less than 1 (Figure 9). When holiday periods are considered, distribution shows apparent outliers. When *R*${}_{0}$ is less than 1, distributions are similar, ranging between 0.75 and 1. Here we notice that asymptomatic individuals and/or unreported cases have played an important role in SARS-CoV-2 transmission, causing some of the unexpected trends (Figure 8). For example, increases in COVID-19 cases occur when the reproducing factor (*R*${}_{0}$) is less than 1.0. This trend occurs because the reproducing factor is a result of data retrieved from official reports and may not represent their real values. According to estimates [34], there is a substantial underreport in cases of COVID-19. Actually, the difference is sometimes one order of magnitude in number of cases, which brings a much higher *R*${}_{0}$ (near to 3).

When the distributions of days of the holiday period and non-holiday are shown (Figure 10), it is seen that the values on holiday days are more concentrated at values *R*${}_{0}>1$. On non-holiday days, *R*${}_{0}$ values are close to 1 but relatively in a range equal to or less than 1.

Despite the distributions shown in Figure 9 and Figure 10, applying the Shapiro–Wilk test [35] to assess whether the distributions are similar to normal distributions, the data that concern holiday periods and *R*${}_{0}$ < 1 have Gaussian behavior by the obtained *p*-value 0.195 (Figure 11c); for all others cases, the *p*-values are 0.000 (Figure 11a,b,d). Because of this, we consider two null hypotheses: (1) H0${}_{p}^{1}$: *R*${}_{0}$ > 1 have the same pattern in holiday periods and in non-holiday periods; (2) H0${}_{p}^{2}$: *R*${}_{0}$ < 1 have same pattern in holiday and non-holiday periods.

Applying the Mann–Whitney test [36], the *p*-values obtained for the null hypothesis were (1) H0${}_{p}^{1}:0.3856$ and (2) H0${}_{p}^{2}:0.0022$, whereas the significance is compared to the value being less than 0.05. The results support the idea that *R*${}_{0}$ > 1 on days that are in holiday periods, differing from days that are not in holiday periods. It also shows us that the *R*${}_{0}$ < 1 does not differ if the day is a holiday period or not.

As a result, these data, in addition to the number of cases, are expected to be important when dealing with disease contamination dynamics. Once we establish a causal relationship between infection rates and the number of confirmed cases, we expect the addition of holiday dates to play an important role in improving the accuracy of predicting COVID-19 deaths.

### 3.2. Principal Components for Mobility Data

We used PCA to transform a set of mobility data variables to an orthogonal mapping, whereas linear regression accounts for the best straight line to fit these data. Figure 12 indicates that a single PC can approximate mobility data variability by 79.19%, whereas two PCs can approximate mobility data variability by 90.25%.

By adopting two PCs, as shown in Table 4, we observed that places such as *retail and recreation*, as well as *transportation terminals*, harm PC${}_{1}$, whereas *residential* had a positive influence on PC. *Parks* had a positive impact on PC${}_{2}$, whereas *workplaces* had a reverse effect. Therefore, with these two PCs, we explained and simplified mobility data variability by 90.25%.

Therefore, the use of PCA helped to convert correlations of six dimensions into community mobility reports into a smaller projection, with 2 dimensions. Figure 13 shows the curves derived using the PCA method, together with the $R_{0}$ value provided by the SEIRD Model.

Figure 14 shows the trend of scattering data, which reveals that PC${}_{1}$ values are more determinant for $R_{0}>1$ during holiday periods when PC${}_{1}$ values are more widely distributed.

Further, we notice that PCA has different behaviors for mobility data for 2020 and 2021. For 2020, it results in more sparse values, which are mainly distributed between −4 and 6, as seen in the PC${}_{1}$ (*X* axis) of Figure 15a (orange dots). For 2021, it is composed, in general, of values between −2 and 1, besides an outlier observed with a value of 4 (Figure 15b).

The proposed hypothesis here is that such PC time-series values can serve as input data for a forecast model for improving prediction. We consider that these patterns presented in the PCs may assist as a behavioral tool for the interpretation and prediction of the COVID-19 daily deaths curve under the influence of these new dimensions generated.

### 3.3. Forecasting COVID-19 Deaths

We suggest considering the extraction of *R*${}_{0}$ and its presented patterns, and the PCs generated, all of them as data input to the LSTM model to predict the number of deaths by COVID-19. The generated PCs represent the community mobility in a smaller number of dimensions, but explain the variability in 90.25% of the community mobility. The set of input configurations that can be used in the proposed network are presented in Table 5.

In order to test and choose for the best configuration, each of them is submitted to a grid-search process of training. In this process, they are executed 50 times and the forecast variations of each model are then analyzed using the test input data. By using the number of cases as the basis for predicting the number of deaths (baseline case), the median value of the results obtained from this method demonstrated that the median of the RMSE error is approximately 214.58 deaths over a 14-day forecast period, as shown in Figure 16a.

Table 6 lists the values for the other model configurations that were submitted to the process in the test-input-data step. As observed in Table 6, forecast was improved by adding the holiday flag to the one-variate model (configuration 2) that initially used only *number of cases* as input data. Another observation was related to the use of principal components as a substitute for the use of *number of cases* as input data. In these cases, we expected a reasonable forecast as an alternative to using only *number of cases* or some other data together with *infected cases*. Adopting only a dimensionless value of $R_{0}$ or $R_{e}$, on the other hand, was not a reasonable approach, especially if one of them was associated with some holiday flags. However, using them as input data in groups and adding holiday flags helped the forecasting problem.

Other configurations, such as using the basic reproduction number (*number of cases* + $R_{0}$) as input data, yielded more accurate results, with an average recorded RMSE of 173.84 (Figure 16b), which was an improvement of 18.99% over the baseline using only *number of cases* as input data. The use of $R_{0}$ and $R_{e}$ with the holiday flag (Figure 16c) produced an RMSE median error of 177.08. This was another configuration that should be highlighted with a higher accuracy of 17.48% over the baseline using only *cases* as input data.

When comparing the generated curves by the model and the COVID-19 daily deaths ground truth (Figure 17), we observed that the $R_{0}$ provided stability in forecasting and reducing the noise in the predicted curve (Figure 17b). However, we noticed that there is a time shift of approximately a week in the predictive curve in certain periods. Considering Figure 17c, generated from the adoption of $R_{0}$, $R_{e}$, and the holiday flag, we observed this time shift problem was solved, despite an increase in RMSE error compared to the previous curve (Figure 17b). Nevertheless, a better and more accurate prediction of deaths caused by COVID-19 was achieved.

## 4. Discussion

Our data suggest that holidays and holiday periods influenced $R_{0}$ values for COVID-19, as well as mobility of people. Non-pharmaceutical interventions, such as social distancing, use of masks, and hand sanitizers were the initial measures to contain the spread of SARS-CoV-2. However, the Brazilian political, economical, and social contexts influenced the establishment of effective public health policies. The lack of extensive social distancing had a major impact on the number of cases and deaths of COVID-19 after holiday periods. The same trend also occurred with activities in which crowding was involved. The earlier social distancing measures are deployed, the sooner that such policies will be relaxed, mainly due to a decrease in cases and deaths by COVID-19 [37].

Several factors help reducing compliance to public health policies, such as socioeconomic inequality, conspiracy-theory-driven noncooperation, and behavioral aspects, such as cognitive bias [38] and free riding [39]. The association of these factors have also influenced disease patterns. Cognitive biases, for example, play an important role in decision making, as they affect individual reasoning, cloud judgment [38], and create anecdotal evidence [40]. A few types of biases affect perception of reality and influence behavior. For example, confirmation bias, i.e., which is the tendency to favor, search for, interpret, and remember information that confirms one’s beliefs [41], has negatively influenced prevention strategies, control measures, and research for COVID-19 [40,42,43]. Additionally, free-riders usually avoid cooperation, exploiting others’ compliance with policies [39]. The lack of compliance usually has a major impact on control measures, as it compromises collective efforts to contain the spread of SARS-CoV-2. Holidays have probably intensified such behavioral factors, causing an increase in the number of cases.

We emphasize that crowding may very well be the main factor for the maintenance of COVID-19 cases and deaths from now on. The circulation of new variants poses a threat to prophylactic measures that are now available against SARS-CoV-2. Although there is evidence that current vaccine strategies are able to elicit an effective immune response against the variants of interest and variants of concern of SARS-CoV-2 described until now, non-pharmaceutical control measures must remain as continuous public policies to avoid the spread of variants that can potentially cause new waves of COVID-19 [44]. Crowding, particularly the crowding that occurred within the first year of COVID-19 in Brazil, rapidly changed COVID-19 reports. The same pattern can be observed on 1 January 2021, when disease reports started increasing after a holiday that traditionally leads to family reunions and festivities.

Nonetheless, part of the lack of compliance with non-pharmaceutical measures may have been validated by the executive branch of the federal government, which has encouraged the population to keep crowding and to avoid the use of masks, at the same time disregarding the severity of SARS-CoV-2 infection and criticizing effective countermeasures deployed by state and municipal authorities all over the country [7]. Unfortunately, control measures were not taken even by federal, state, and municipal authorities. The direct result of the non-compliance with countermeasures is a mean mortality rate of 2.345, which is approximately 77% higher than the rest of the world [45]. Additionally, COVID-19 denial has also been widely reported on traditional networks that support the Federal Executive Branch policies, on unchecked digital media, and on social networks, adding mistrust in science into this scenario. As a result, part of the population internalized political and economical biases that were part of the federal government’s agenda instead of complying with collective control measures, which resulted in one of the most severe examples of COVID-19 problems in the world [7].

Non-compliance with COVID-19 collective control measures is also related to socioeconomic status in the population in several countries. Unfortunately, informal employment is a worldwide tendency, and Brazilian reality is not different from the rest of the world. In fact, issues that have historically present in the Brazilian territory were magnified with the sanitary scenario imposed by COVID-19. Increases in several types of violence, poverty, and differential access to health and education are among the hurdles Brazilians have to deal on a daily basis. Festivities and hallmark holidays are usually a chance for informal workers to increase individual or family income as they increase the demand for consumer goods such as food, clothes, and beverages. Public policies to control the spread of SARS-CoV-2, such as lockdowns and social distancing, caused a major impact in families whose income relies on informal jobs and social interactions. Informal employment causes significant tax loss and reduced public revenues, leading to less available resources for important public services, such as social protection and health care in Brazil, but also contributes to poorer working conditions and unfair competition for legitimate businesses and collective bargaining [46]. Data from Bahia, a state from the northeast region of Brazil, indicates that withdrawal of informal workers decreases the productive capacity of the country and leads to pronounced negative impacts on the economy, mainly in the service sectors. However, federal, state, and regional programs for income transfer helps to mitigate the negative impacts of COVID-19 by 50% [47]. Implementation of control measures associated with income transfer policies would not only influence mobility of people but also have a positive impact in COVID-19 reports.

## 5. Conclusions

As a result of the experiments, we unveil some intriguing concerns. First, we have discovered that the effects of holidays, or holiday periods, cause immediate increases in $R_{0}$ and $R_{e}$ trends. These factors, extracted from the SEIRD model, have a significant impact on the COVID-19 death curves and reports.

Furthermore, principal components generated with community mobility report data using the PCA approach indicate that holidays may cause distinct report patterns over time, which can be analyzed to improve COVID-19 regression of death curves. Furthermore, the PCA approach has proven to be important because it reduces the dimensionality of the feature space, and with fewer input dimensions, the model is easier to find.

Cases combined with holiday periods, cases in association with $R_{0}$, or holiday periods with $R_{0}$ produced an effective strategy of analysis compared to using the current number of cases to predict future deaths. Furthermore, when access to epidemiological data is limited, the use of community mobility is a promising alternative as it produced better results than cases as input data.

Based on the trained model, it is possible to generate synthetic data, for example, simulating an increase in mobility, and see how this affects the model’s forecast. If the trained model has a high forecast accuracy, we can confidently estimate the impact of each feature on the dynamic of the death curve. This finding may be used to help authorities managing resources in the event of future epidemics.

Besides having produced important results showing that we are in the right research direction, our current data are not final, and further work should be done in order to draw a thorough and final conclusion on the enhancement of the models for prediction. For that purpose, it will be necessary in future work to rank-order factors (in order of relevance), taking into account the literature consensus on factors in COVID-19 infection rate. Some papers list more than 50 potential features for predicting the number of cases, including mobility and climate variables such as temperature, humidity, and air pollution [48]. To predict deaths, the number of vaccinations, population age, and the number of available ICUs should all be considered.

Finally, using a large number of variables as input data will not necessarily improve the prediction of COVID-19 fatalities in an LSTM model. However, once a large number of data have been collected, it is possible to use exploratory methods to conduct trials that contribute to and improve the accuracy of the model. We also intend to adopt other forms of data visualization and approaches that can improve understanding of the virus’s spread dynamics and help make predictions. A possible approach is the use of complex networks, which help to understand how variables interact, as shown in [49].

In the near future, we intend to study and add new variables to our prediction model, such as the association and influence of countermeasures defined by the federal, state, and municipal governments and particularities of people from different regions of the country, among others. With this approach, a comparison between the effectiveness of control measures by different states or municipalities can be evaluated to help avoid similar scenarios in the future.

## Figures and Tables

**Figure 1 ijerph-18-11595-f001:**
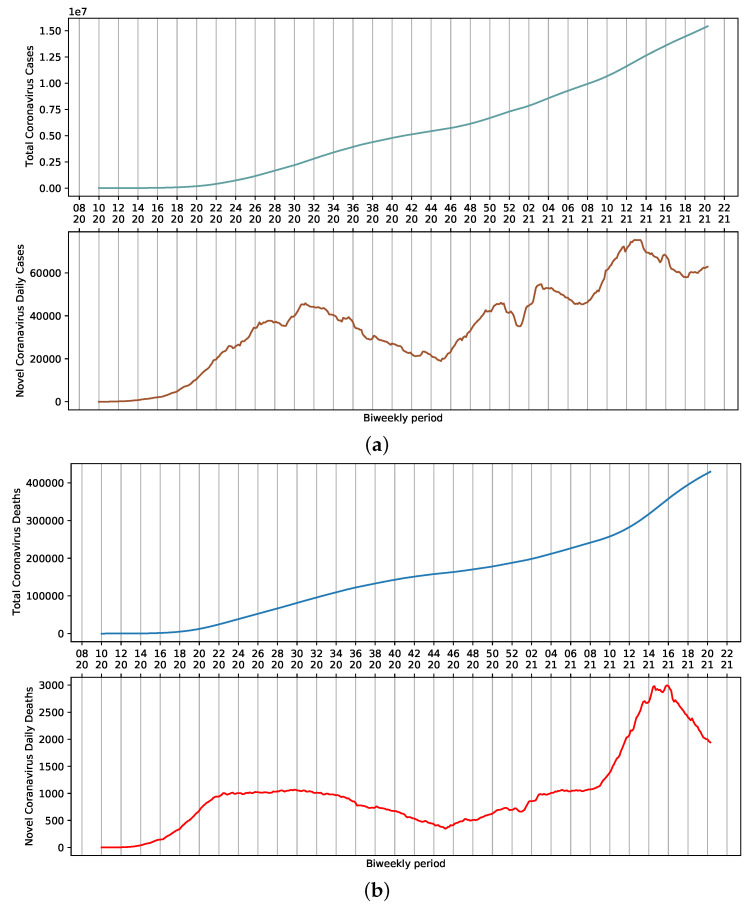
Moving average for COVID-19 in Brazil, presented according to epidemiological week. (**a**) Cumulative and daily case reports. (**b**) Cumulative and daily deaths.

**Figure 2 ijerph-18-11595-f002:**
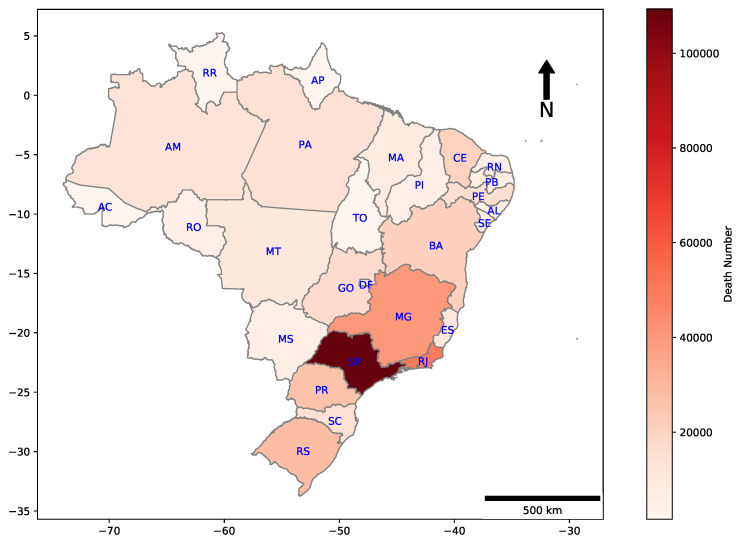
Heat map of cumulative deaths by COVID-19 in each Brazilian state. E.g., São Paulo, which is the most populated Brazilian state, had 109,241 deaths until the date that this research has been finished.

**Figure 3 ijerph-18-11595-f003:**
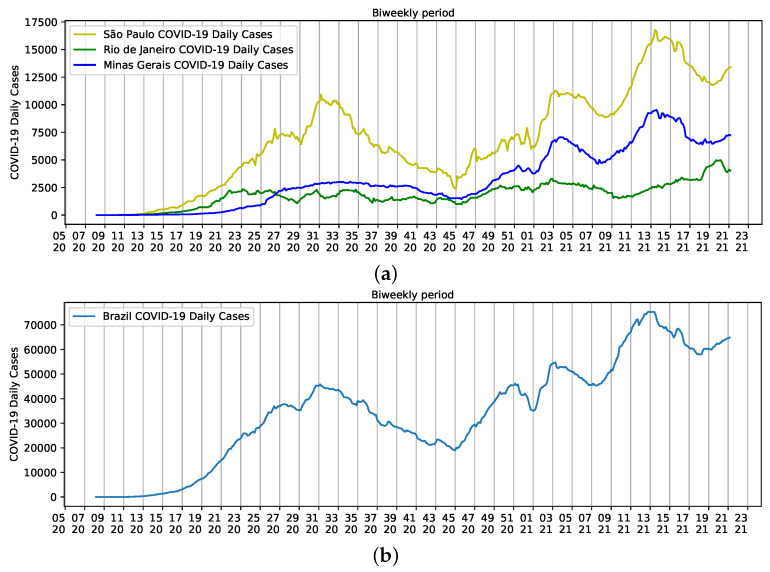
Biweekly cases of COVID-19 reported from epidemiological week five of 2020 to epidemiological week 22 of 2021. (**a**) States of São Paulo (yellow), Rio de Janeiro (Green), and Minas Gerais (Blue); (**b**) Brazil.

**Figure 4 ijerph-18-11595-f004:**
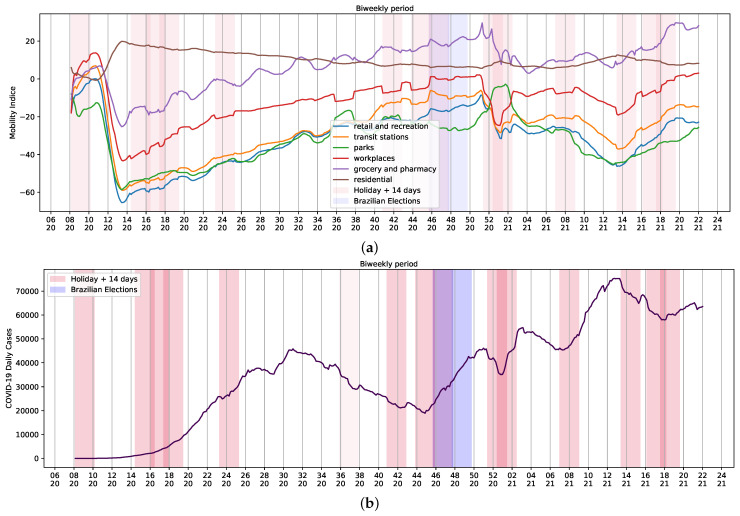
Brazilian Community Mobility Reports for retail and recreation, transit stations, parks, workplace, residential and grocery, and pharmacy during ordinary days and national holidays during the COVID-19 pandemic. (**a**) Community mobility reports and holidays; (**b**) biweekly reports of COVID-19 and holidays. Red represents the holiday plus 14 days; purple represents Brazilian elections. Mobility data provided by Google LLC.

**Figure 5 ijerph-18-11595-f005:**
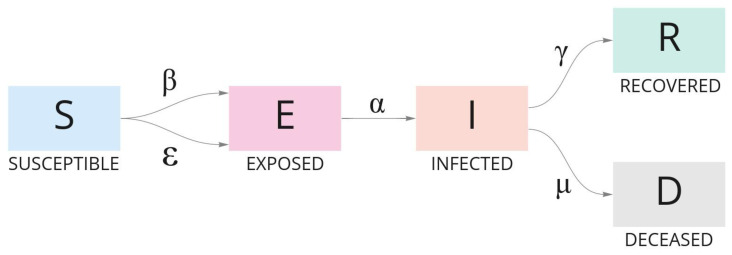
The evolution of individuals in a population during the course of an outbreak according to the SEIRD model. Each group of individuals is defined by an ODE and categorized as Susceptible, Exposed, Infected, Recovered, and Dead.

**Figure 6 ijerph-18-11595-f006:**
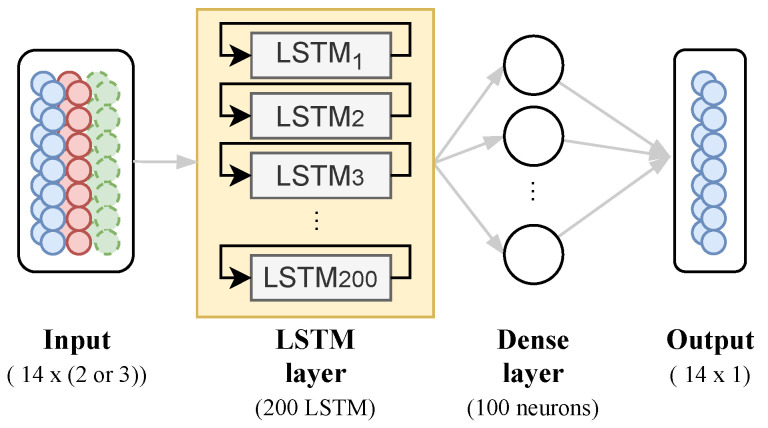
Neural Network Architecture proposed.

**Figure 7 ijerph-18-11595-f007:**
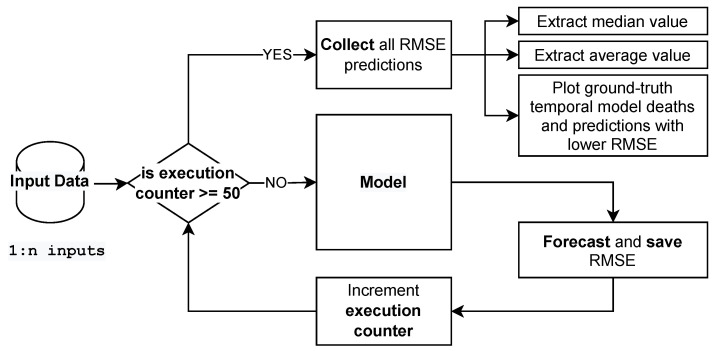
Workflow applied to grid search of input data sets.

**Figure 8 ijerph-18-11595-f008:**
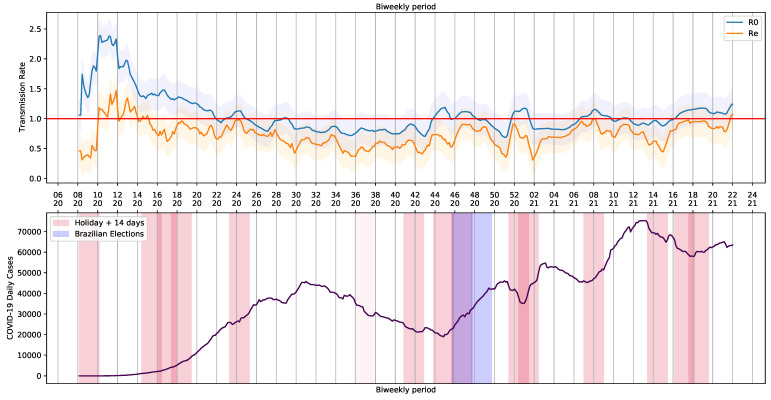
Basic Reproduction Number $R_{0}$ and COVID-19 Daily Cases in Brazil.

**Figure 9 ijerph-18-11595-f009:**
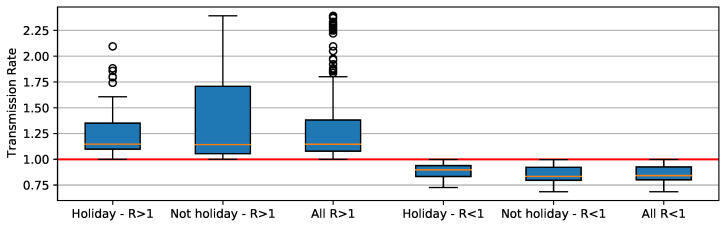
Boxplot distribution of $R_{0}$ value for holidays and non-holidays.

**Figure 10 ijerph-18-11595-f010:**
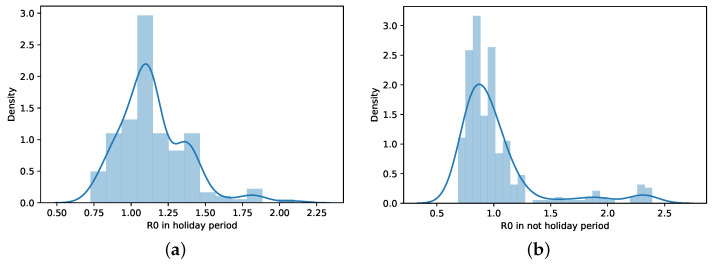
Histogram and distribution of *R*${}_{0}$ in holiday and non-holiday periods. (**a**) *R*${}_{0}$ distribution—holiday period. (**b**) *R*${}_{0}$ distribution—non-holiday period.

**Figure 11 ijerph-18-11595-f011:**
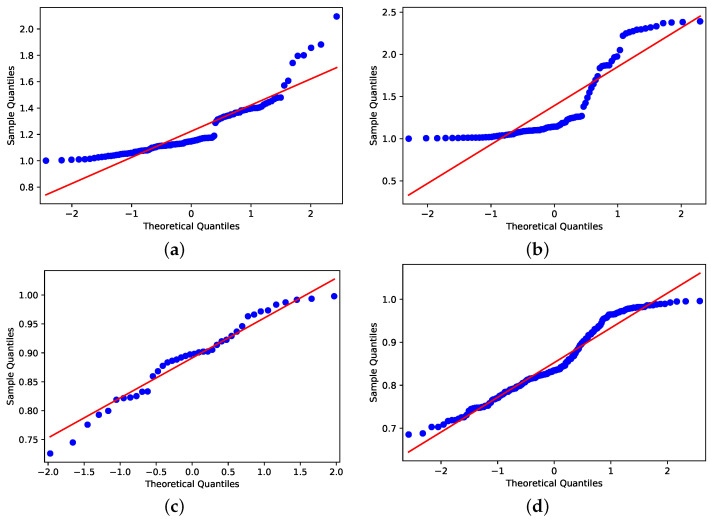
Shapiro–Wilk outcomes. (**a**) Holiday period with *R*${}_{0}$ > 1. (**b**) Not holiday period with *R*${}_{0}$ > 1. (**c**) Holiday period with *R*${}_{0}$ < 1. (**d**) Not holiday period with *R*${}_{0}$ < 1.

**Figure 12 ijerph-18-11595-f012:**
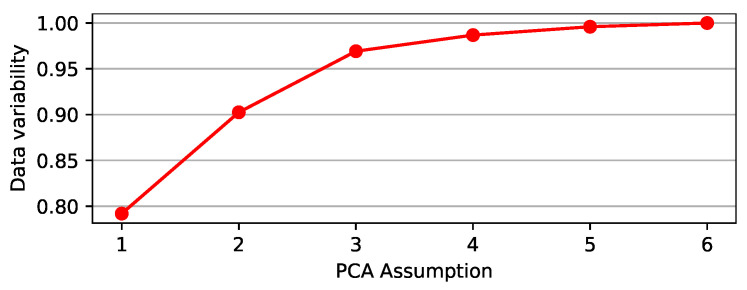
Percentage variability of the mobility reports data, which may be represented by PCA.

**Figure 13 ijerph-18-11595-f013:**
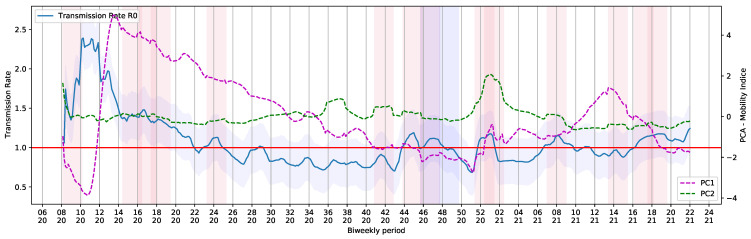
PC${}_{1}$ and PC${}_{2}$ shown with Basic Reproduction Number $R_{0}$ during the COVID-19 timeline in Brazil.

**Figure 14 ijerph-18-11595-f014:**
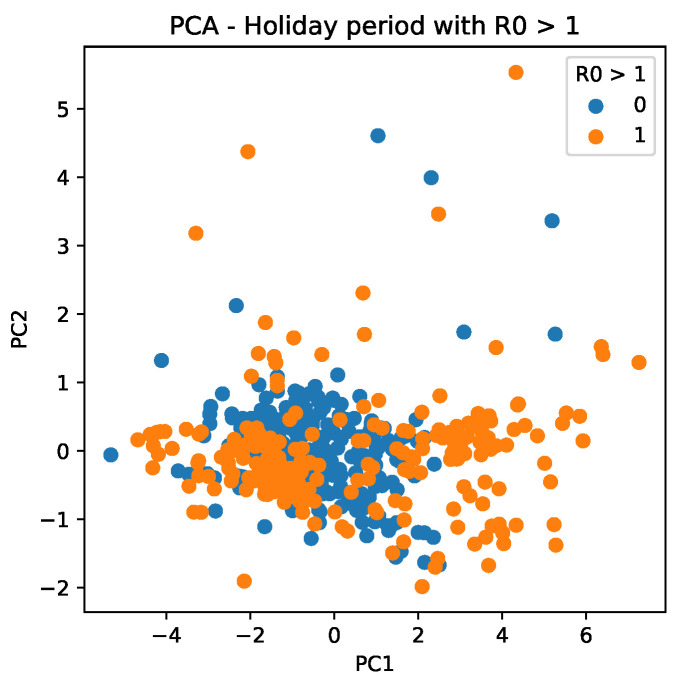
PC scattering based on mobility reports considering $R_{0}>1$ and the holiday period. A point with a value of 0 indicates that it is not a holiday, whereas a value of 1 indicates that it is a holiday period.

**Figure 15 ijerph-18-11595-f015:**
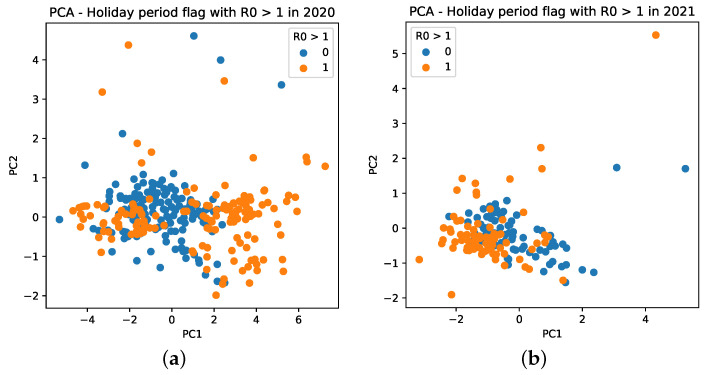
Cluster based on scattering of PC values using mobility data during the COVID-19 epidemic in Brazil. A point with a value of 0 indicates that it is not a holiday, whereas a value of 1 indicates that it is a holiday period. (**a**) PC scattering based on mobility reports considering $R_{0}>1$ and holiday period in 2020. (**b**) PC scattering based on mobility reports considering $R_{0}>1$ and holiday period in 2021.

**Figure 16 ijerph-18-11595-f016:**
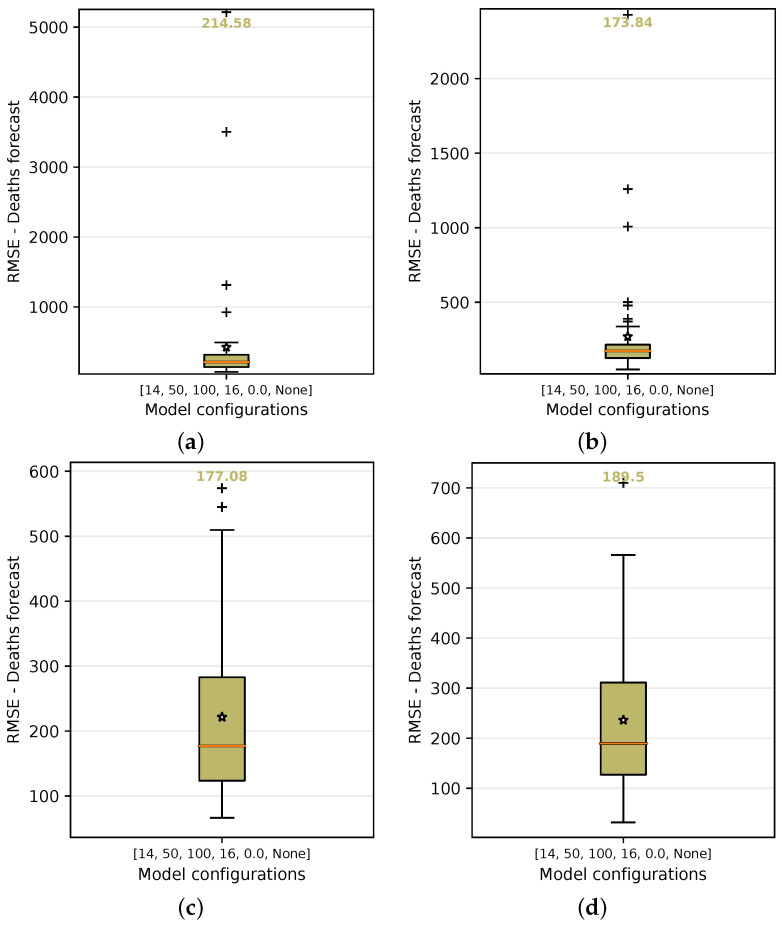
RMSE median and average error in the boxplot on grid search using the following as input data: Cases, Cases + $R_{0}$, $R_{0}$ + $R_{e}$ + Holiday flag, and PC${}_{1}$ + PC${}_{2}$. For each configuration, the graphs show the median and average of the accumulated error in predicting deaths over 50 repeated experiments. (**a**) Cases. (**b**) Cases + $R_{0}$. (**c**) $R_{0}$ + $R_{e}$ + Holiday flag. (**d**) PC${}_{1}$ + PC${}_{2}$.

**Figure 17 ijerph-18-11595-f017:**
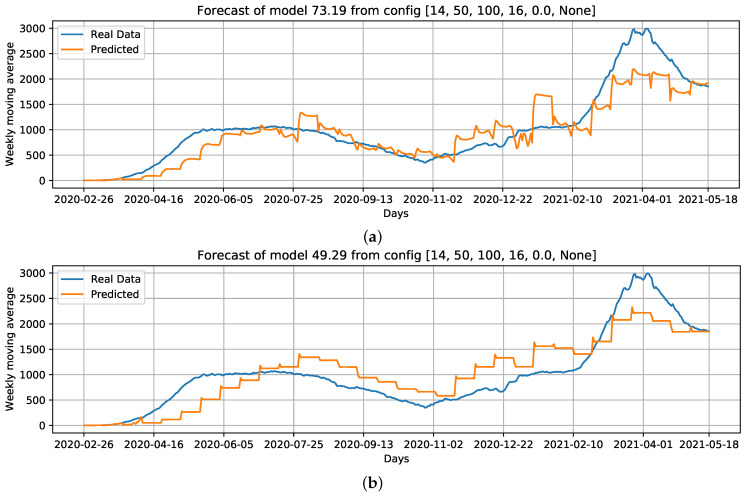
Curves of selected settings: **Conf.1**: Cases, that is, our baseline case; considering which parameters can improve the baseline case, we select **Conf.3**: Cases + $R_{0}$; without using Cases as an input feature, we select **Conf.9**: $R_{0}$ + $R_{e}$ + Holiday flag, and; **Conf.10**: PC${}_{1}$ + PC${}_{2}$ as input data. For these configurations, we draw the best curves (with the lowest RMSE) of each configuration over the test data. (**a**) COVID-19 daily deaths forecast using cases as input data. (**b**) COVID-19 daily deaths forecasts using Cases + $R_{0}$ as input data. (**c**) COVID-19 daily deaths forecasts using $R_{0}$ + $R_{e}$ + Holiday flag as input data. (**d**) COVID-19 daily deaths forecasts using PC${}_{1}$ + PC${}_{2}$ as input data.

**Table 1 ijerph-18-11595-t001:** List of all column names contained in the open-access dataset of COVID-19 [17].

Column Name	Column Name
epi week	total cases per 100 k inhabitants
date	deaths by totalCases
country	recovered
state	suspects
city	tests
newDeaths	tests per 100 k inhabitants
deaths	vaccinated
newCases	vaccinated per 100 k inhabitants
totalCases	vaccinated second
deathsMS	vaccinated second per 100 k inhabitants
totalCasesMS	vaccinated single
deaths per 100 k inhabitants	vaccinated single per 100 k inhabitants

**Table 2 ijerph-18-11595-t002:** Brazilian Holidays. * According to Regulamentation N${}^{\circ }$ 2.621/GM-MD, published on 5 August [19], the Brazilian Ministry of Defense prohibited the Army Forces from participating in commemorative events that could result in crowding, such as parades and military demonstrations.

**Date**	1 Jan	24 Feb	25 Feb
**Weekday**	Wed	Mon	Tue
**Holiday**	New Year’s Day	Carnival	Carnival
**Date**	10 Apr	21 Apr	1 May
**Weekday**	Fri	Tue	Fri
**Holiday**	Good Friday	Minas Conspirancy	International Workers’
**Date**	11 Jun	7 Sep	12 Oct
**Weekday**	Thu	Mon	Mon
**Holiday**	Corpus Christi	Independence day *	Our Lady of Aparecida
**Date**	2 Nov	15 Nov	25 Dec
**Weekday**	Mon	Sun	Fri
**Holiday**	All Souls’	Proclamation of the Republic	Christmas

**Table 3 ijerph-18-11595-t003:** Distribution of $R_{0}$ value for holidays and non holidays over the period of 448 days.

	$R_{0}>1$	$R_{0}\leq 1$
holiday period	80 days	60 days
not a holiday period	104 days	204 days

**Table 4 ijerph-18-11595-t004:** Influence of mobility reporting data on PC projection variability.

	PC-1	PC-2
retail and recreation	−0.441861	0.149809
grocery and pharmacy	−0.378593	−0.052756
parks	−0.341852	0.790273
transit stations	−0.450280	−0.063395
workplaces	−0.394821	−0.570140
residential	0.431192	0.145478

**Table 5 ijerph-18-11595-t005:** All settings considered for the grid-searching, evaluating $R_{0}$ and $R_{e}$, extracted from SEIRD Model, and Holiday flags, which is a flag value representing the holiday + 14 days subsequent, and PCs generated by the PCA method. For all configurations it proposed is to forecast the COVID-19 daily deaths for the last bi-weekly period, reserved for testing.

Configuration Number	Input Data
1	Cases
2	Cases + Holiday flag
3	Cases + $R_{0}$
4	$R_{0}$
5	$R_{e}$
6	$R_{0}$ + $R_{e}$
7	$R_{0}$ + Holiday flag
8	$R_{e}$ + Holiday flag
9	$R_{0}$ + $R_{e}$ + Holiday flag
10	PC${}_{1}$ + PC${}_{2}$
11	PC${}_{1}$ + PC${}_{2}$ + $R_{0}$
12	PC${}_{1}$ + PC${}_{2}$ + $R_{0}$ + Holiday flag

**Table 6 ijerph-18-11595-t006:** RMSE over test input data median and average prediction error of 50 models for each configuration described in Table 5.

Configuration	1	2	3	4	5	6
Median error	214.58	183.11	173.84	219.05	221.37	191.65
Configuration	7	8	9	10	11	12
Median error	216.11	223.04	177.08	189.05	199.49	209.01

## Data Availability

Code and data used in this research can be found in https://github.com/Natalnet/ncovid-holidays-paper (acessed on 30 October 2021).

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
