# Peer review of "National Holidays and Social Mobility Behaviors: Alternatives for Forecasting COVID-19 Deaths in Brazil"

_ijerph, 2021, doi:10.3390/ijerph182111595_

Round 1

Reviewer 1 Report

The paper entitled “National Holidays and Social Mobility Behaviors: Alternatives for Forecasting COVID-19 Deaths in Brazil by  Dunfrey Pires Aragão et al. investigates the influence of holidays and community mobility on the transmission rate and death count of COVID-19 in Brazil. The work is very interesting and inovative and it should be published. However there are some points to be taken into account by the authors.  

  1. In Figure 1 the authors have to explain why the curve of deaths does not reflects the expansion of Covid 19. For example in the period from 22 to 34 weeks (Fig.1.b) deaths are stable whereas in the same time Covid 19 cases increase.
  2. In Figure 2 it is obvious that after holidays cases of Covid 19 generally increase. However it is not clear the same behavior in the period from 14 to 20 week of 2020.
  3. In Figure 3 why the most populated area of Rio de Janeiro does not present the  more Covid 19 cases than Sao Paulo?
  4. In Figure 7 the authors have to explain how Covid 19 cases increase when the reproducing factor is less than 1.0?

Finally it could be better the authors to analyze their data according other works such as: Modeling and Forecasting the COVID-19 Temporal Spread in Greece: An Exploratory Approach Based on Complex Network Defined Splines” by K.Dermetzis , June 2020, International Journal of Environmental Research and Public Health 17(13):4693

Author Response

Please, see attached file.

Reviewer 2 Report

National Holidays and Social Mobility Behaviors: Alternatives for Forecasting COVID-19 Deaths in Brazil

The objective of this paper is to determine the impact of holydays in the contagious process of COVID-19. The method of study is applied over data from Brazil. To develop their study, the authors combine several techniques to pursue conclusions regarding the study’s objective.                                 

General Comments

The manuscript is grammatically well written. However, the highly complex method presented makes the document hard to read. The authors begin setting up a contagious model they call SEIRD. In this model the letters E and D, representing the Exposed and Deceased, are added to the conventional SIR model. These status additions to the model are hardly justifiable.  In the conventional SIR model the latter R stands for Retired, thus including recovered and deceased. According to the present study the Recovered get immunity and thus cannot participate in further contagious process. Just as the Deceased cannot either. The condition of Exposed seems also as an unnecessary refinement of SIR model. I find it very difficult to differentiate between Exposed and Infected. I simply don’t see the difference in the rolls of the two conditions. Moreover, there seem to be two different processes to go from S to E. This makes hard, if not impossible, to reach a steady convergent value for Beta and Epsilon. Finally, Introducing E and D makes the model more complex and difficult to keep under control and compromising the clarity of the experiment.

There may be nothing wrong with the method followed to estimate Ro. But all this formulation looks unnecessarily cumbersome. Similarly, it I not clear at all the need for applying Neural Nets in this situation. The remaining of the document shows several technologies used for purposes very difficult to follow.   

Final Statement.

This study has a valid objective as is to determine the influence of holydays in the COVID-19 contagious process. The writing is fine but in my opinion the document needs a better organization. I think the strategy developed to compute the numbers sought, is exaggeratedly complex, and the authors themselves have problems explaining it. The result is that the method did not allow them to show clear conclusions. Only general phrases integrate this section of the manuscript.

Author Response

Please, see attached file.

Reviewer 3 Report

I think the paper is interesting and has a lot of potential but I think some major improvements need to be made particularly in framing the goals of the paper the experiments and the results.  

  • When the goals of the paper are initially given in the introduction they feel a bit disjointed as if they don't relate to each other. However, I think they do relate and are interconnected. From what I got out of the paper, the SEIRD model is done for two reasons 1) to do a basic check that we see differences in R during holiday periods and 2) to use R as an input to the PCA analysis. The PCA analysis is done to reduce the number of inputs to the LSTM model and the LSTM model is done to predict COVID-19 deaths.  Thus I think the actual goal of the paper is to predict COVID-19 daily deaths in Brazil taking into account mobility data and changes in mobility due to holidays using the other models as input.  
  • For the SEIRD model and the PCA I think it needs to be better emphasized that they are both being done to be used as inputs to the LSTM model. Yes there are other things that are learned from these analyses but for the flow of the paper they are inputs to the LSTM model. Otherwise the paper is just three somewhat connected experiments
  • I also think it might be helpful for there to be a statistical test showing the differences in R0 between holiday and non holiday times (Table 3) which would help to justify the whole work and using R in the LSTM model
  • I think the PCA analysis is really interesting but I think to be included in the paper it needs a couple of things. 
    • More justification. The authors state that the PCA is done to reduce the dimensions of the data but why do the dimensions need to be reduced? Is there a limitation of the number of variables that can be used in an LSTM model? As this seems to be a big part of this work the authors need to provide more explanation of why it is done. 
    • More explanation of the results. The authors briefly discuss the PCA results but I think this should be done in a little more detail to give the reader an idea of how they will impact the model. This is kind of started in lines 275 - 279 but if the authors could provide more interpretation of the PCs. For example, it seems like PC1 might be capturing the differences between people being out in the community and moving between regions versus at home.
  •  For the section on the LSTM model and forecasting deaths I think this should be the main results of the paper and also need some more detail. 
    • Which set of factors do the authors think are the best? Which should others choose if they were following the same methods?
    • Also it might help if the authors discuss why predicting deaths over the next 14 days is important. Predicting deaths 14 days ahead with current data won't stop the deaths as those but might help to inform the healthcare system of future short term demand. This is important in justifying why you are doing this study. Or is the study more to look at why factors and combination of factors impact deaths if this is the case this should be reflected in the goals in the introduction.  
  • I think the conclusion needs to relate back to the work a little bit more. I'm not sure I see that "From our results, we can observe that the non-adoption of social distance has an impact on the incidence of cases and deaths after holiday periods, also noticed on other activities where crowding has occurred. Nonetheless, it is observed that early deployment of social distancing measures can result in early flexibility of these same policies, reflecting times when epidemiologically there may be a rising increase in cases and deaths."  
    • What I see is that the LSTM model can predict deaths in 14 days based off of mobility, R and current cases. What do the combinations of factors that are included in the better LSTM models tell the authors about what impacts deaths? How can we use these results to better prepare for the next pandemic or the next wave of the COVID-19 pandemic.  Why is it important to combine factors with a PCA analysis?

I think the work does have a lot of potential I just think the presentation, details and justification need to be reconsidered and expanded on. I think the PCA part is really interesting on its own as an analysis of how the mobility patterns are related and as an input to the model but more explanation is needed. 

Author Response

Please, see attached file.

Round 2

Reviewer 2 Report

Important improvements are noticeable. The manuscript is now very interesting and nicely presented. Conclusions are relevant and well sustained. It deserves publication now.

Author Response

Reviewer 1:

Q1: Important improvements are noticeable. The manuscript is now very interesting and nicely presented. Conclusions are relevant and well sustained. It deserves publication now.

A1: Thanks for your comments. We have done a final minor revision checking the manuscript for small issues.

Reviewer 3 Report

I'm happy with the authors changes. I think they have addressed all of my concerns. There are some references that should be checked, in the discussion there are some that latex does not appear to recognize that show up as a ? 

Author Response

Reviewer 2:

Q2: I'm happy with the author's changes. I think they have addressed all of my concerns. There are some references that should be checked, in the discussion, there are some that latex does not appear to recognize that show up as a ? 

A2: Thanks for your comments and suggestions. We have done a last minor revision of the entire manuscript. Actually, we had already detected those missing references just after resubmission of the last version and it is corrected in this version. We have also done a language review and spell check, hopefully, we removed all problems.